# High Carbohydrate Diet Is Associated with Severe Clinical Indicators, but Not with Nutrition Knowledge Score in Patients with Multiple Myeloma

**DOI:** 10.3390/ijerph18105444

**Published:** 2021-05-19

**Authors:** Ema Borsi, Costela Lacrimioara Serban, Cristina Potre, Ovidiu Potre, Salomeia Putnoky, Miruna Samfireag, Raluca Tudor, Ioana Ionita, Hortensia Ionita

**Affiliations:** 1Discipline of Hematology, Department of Internal Medicine, Victor Babes University of Medicine and Pharmacy, 300041 Timisoara, Timis County, Romania; borsi.ema@umft.ro (E.B.); potre.cristina@umft.ro (C.P.); potre.ovidiu@umft.ro (O.P.); ionita.ioana@umft.ro (I.I.); ionitahortensia@gmail.com (H.I.); 2Department of Functional Science, Victor Babes University of Medicine and Pharmacy, 300041 Timisoara, Timis County, Romania; 3Hematology Clinic, Timisoara’s Emergency City Hospital, 300723 Timisoara, Timis County, Romania; samfireag.miruna@umft.ro; 4Department of Microbiology, Victor Babes University of Medicine and Pharmacy, 300041 Timisoara, Timis County, Romania; putnoky.salomeia@umft.ro; 5Department of Internal Medicine, Discipline of Clinical Practical Skills, Victor Babes University of Medicine and Pharmacy, 300041 Timisoara, Timis County, Romania; 6Department of Neurology, Victor Babes University of Medicine and Pharmacy, 300041 Timisoara, Timis County, Romania; tudor.raluca@umft.ro

**Keywords:** multiple myeloma, nutrition knowledge, nutrients, diet, clinical outcome

## Abstract

Although the survival rate of patients diagnosed with multiple myeloma has doubled over the last few decades, due to the introduction of new therapeutic lines and improvement of care, other potential contributors to the therapeutic response/relapse of disease, such as nutrient intake, along with nutrition knowledge, have not been assessed during the course of the disease. The purpose of this research was to assess nutrition knowledge and diet quality in a group of patients with a diagnosis of multiple myeloma. Anthropometric, clinical and biological assessments and skeletal survey evaluations, along with the assessment of nutritional intake and general nutrition knowledge, were performed on 61 patients with a current diagnosis of multiple myeloma. A low carbohydrate diet score was computed, classified in tertiles, and used as a factor in the analysis. Patients in tertiles indicative of high carbohydrate or low carbohydrate intake showed significant alteration of clinical parameters, such as hemoglobin, uric acid, albumin, total proteins, beta-2 microglobulin, percentage of plasmacytes in the bone marrow and D-dimers, compared to patients in the medium carbohydrate intake tertile. Nutrition knowledge was not associated with clinical indicators of disease status, nor with patterns of nutrient intake. Better knowledge of food types and nutritional value of foods, along with personalized nutritional advice, could encourage patients with MM to make healthier decisions that might extend survival.

## 1. Introduction

In 2020, the global age-standardized incidence rates for multiple myeloma (MM) were 2.2/100,000 for males and 1.5/100,000 for females, with an age-standardized mortality rate of 1.1/100,000 [1]. Recognized and relatively constant risk factors are older age, male gender and African ethnicity [2]. Due to new therapies and constant improvements in care, survival has more than doubled over recent decades. Recently, Usmani et al. [3] recognized that in newly diagnosed transplant eligible MM patients, age (<65 years), non-IgA isotype, normal albumin levels, low beta-2 microglobulin ≤ 3.5 mg/dl, serum creatinine levels < 2 mg/dL, hemoglobin levels ≥10 g/dL and platelet count ≥ 150 k/μL had a positive effect on the 10-year survival, but elevated serum LDH levels and any cytogenetic abnormalities did not negatively predict 10-year survival. Still, the survival of patients with MM is impaired, with only 10–15% reaching the lifespan of the general population [3].

Recently, the relationship between diet and the risk of MM, as well as the relationship between diet and prognosis of patients with MM, has been tackled in several publications; however, most of the relationships that were studied pertained to pre-diagnosis diet. Protective factors for MM in adulthood included increased intake of fruits (over three servings per week) [4], increased intake of fish [5] and current alcohol intake in both genders compared to non-drinkers [6]. Risk factors for MM included the intake of more than one serving of artificially sweetened drinks [7]. Lee et al. [8] determined that an unhealthy overall diet pre-diagnosis is associated with poor survival rates in patients with MM.

Although the relationship between nutrition knowledge, food choices and food intake is complex, little is known about the level of nutrition knowledge in patients diagnosed with MM and the quality of their diets after diagnosis. Our aim was to assess the nutritional knowledge and quality of diets in a cohort of patients with MM, and prepare an intervention study to increase general and specific nutrition knowledge, including nutrition counseling.

## 2. Materials and Methods

### 2.1. Participants and Samples

From a cohort of 76 patients followed up in the Hematology Department of the Municipal Emergency Hospital, Timisoara, Romania, 61 participants were included in this research (Figure 1). Patients were recruited between September 2020 and January 2021, during the second wave of the COVID-19 pandemic. This research was designed as a cross-sectional investigation pilot study. Nevertheless, the sample size was sufficient, taking into account a minimum sample of 59 participants, calculated by Viechtbauer et al. [9] for pilot studies. 

Diagnosis, staging and therapy were established using ESMO clinical practice guidelines [10]. Data were collected during follow-up visits, and included anthropometric, hematological, biochemical and immunological parameters, radiological data, whole body MRI and type of and response to treatment, as well as survival duration in months (from the time of diagnosis up to the end of the study). For this pilot study, the following inclusion criteria were used: age >18 years, diagnosis of MM in clinical stage 1–3 and a follow-up of at least 6 months, according to the protocol. For the purposes of analysis, the exclusion criteria were set as follows: allogeneic SCT, monoclonal gammopathy of undetermined significance and drug-naïve patients. The exclusion criteria also took into consideration the inability to provide informed consent, and the inability to provide accurate anamnestic data. 

Patients were included only after signing the informed consent. The entire study was conducted according to the principles stated in the Declaration of Helsinki, and was approved by the Ethics Committee of the “Victor Babes” University of Medicine and Pharmacy, Timisoara, Romania, with no. 6/2019.

### 2.2. Clinical and Biological Evaluation

Height and weight were measured as per the international guidelines, and were used for the calculation of body mass index (BMI; BMI = weight (kg)/height (m)^2^). The nutritional status of the participants was determined using the following BMI thresholds: underweight (BMI below 18.5 kg/m^2^), normal weight (18.5–24.9 kg/m^2^), overweight (25–29.9 kg/m^2^) and obese (over 30 kg/m^2^). All patients underwent whole-body low-dose computed tomography. All blood samples were collected in the morning. Complete blood count implied the collection of venous blood in a sterile EDTA vacutainer, using fluorescence flow cytometry. Coagulation tests consisted of the following: dosing of fibrinogen through a coagulometric method, and performing quantitative tests to dose D-dimer; analysis of venous blood collected in a vacutainer with sodium citrate, with plasma separation by centrifugation, and processing using a latex method through automatic agglutination with photometric detection. The samples for serum blood urea nitrogen, creatinine, calcium and LDH testing were collected in a sterile vacutainer without anticoagulant, with/without gel separator, by a spectrophotometric technique. Beta-2 microglobulin was assessed using the nephelometric method. The samples underwent analysis via serum protein electrophoresis with immunofixation, serum-free light chain quantification, heavy/light chain quantification, immunoglobulin and total proteins. All patients underwent bone marrow aspiration. 

The diagnosis of chronic kidney disease was established by measuring serum creatinine levels, in order to calculate the glomerular filtration rate, and by measuring the urinary albumin/creatinine ratio, in order to detect proteinuria [11]. Osteoporosis was diagnosed based on bone mineral density determination obtained from dual energy X-ray absorptiometry assessment [12]. Major depression was diagnosed by a psychiatrist, using the DSM-5 criteria [13], and clinical assessments were conducted in order to diagnose any potential peripheral neuropathy. Information about diarrhea/constipation was obtained during the anamnestic process, and information about any infections was extracted from clinical records. 

### 2.3. Nutritional Assessment 

Dietary intake assessment was performed using a validated FFQ, consisting of 53 food items; it investigated food intake during the last 30 days [14,15]. For each item, the frequency and the usual amount of consumed food items were investigated. To give an estimate of the quantity of fat or added sugar, additional questions were asked in relation to some of the items. The intakes were converted to grams, in accordance with the guidelines of household scales [16]. Using a computer program specially created for this purpose, the energy and macronutrient intakes were calculated for each individual. Macronutrients were transformed into a percentage of contribution to total energy, and then the adequacy of intake was calculated as per the European Food and Safety Authority (EFSA) recommendations [17]. “Inadequate intake” had different meanings for different macronutrients: for proteins, “inadequate” meant lower than the recommended values; for fat, “inadequate” represented only the values above the upper recommended threshold; as for carbohydrates, the values were both lower and upper thresholds. The energy-adjusted values of intake were also computed for saturated fat, fibers and alcohol. 

A low carbohydrate diet (LCD) score was computed using deciles of the percentage of macronutrients. For carbohydrates, the lowest decile received a score of 10, and the highest decile received a score of one. For fat and protein, the lowest decile received a score of one, and the highest received a score of 10. All individual macronutrient scores were added to obtain the LCD score, which ranged from a possible lowest score of 3 to a highest score of 30, with higher scores indicating higher adherence to a low carbohydrate diet [18,19]. The LCD score was transformed into tertiles and further used as a three-layer factor in statistical analysis. 

### 2.4. Nutrition Knowledge Assessment

A previously validated questionnaire [20] was used to assess nutrition knowledge. The nutritional knowledge assessment included 88 items. Each item allowed for only one answer, and the obtained points were added for each correct answer provided by the responders, in order to determine the sub-score per section and the total knowledge score. 

### 2.5. Statistical Analysis

Statistical analysis was performed using the Statistical Package for Social Sciences (SPSS Corp, version 18, Chicago, IL, USA). Frequency and percentage were used for the description of categorical data, and mean and standard deviation were used for continuous data. The normality of data was evaluated by the Shapiro–Wilk test. All continuous variables were normally distributed, therefore parametric tests (*t*-test and ANOVA) were employed for the comparisons of means between the categories. Chi-square or Mann–Whitney or Kruskal–Wallis tests were used to compare the characteristics of participants for categorical data.

## 3. Results

Within the group, 13.1% (8) of participants were in disease stage 1, 21.3% (13) were in disease stage 2 and 65.6% (40) were in disease stage 3. Females represented 57.4%. Mean age was 65.2 +/− 9.5 years, within the age range of 41–84 years. The mean BMI was 26.2 +/− 4.2 kg/m^2^, with 57.4% (35) being overweight and obese, and the remaining 42.6% (26) having a weight within normal limits. In Table 1, the clinical features of the cohort of patients diagnosed with MM are presented using disease stage as a factor. The proportions of genders, kappa or lambda light chains in blood and urine, the presence of infections, increased viscosity, depression and therapeutic response did not statistically differ between stages 1 and 2 versus stage 3. Additionally, mean age, BMI and number of therapeutic lines were not statistically different between the two categories. All the rest of the clinical variables showed statistical differences between stages 1 and 2 versus stage 3. 

Unadjusted energy (kcal), macronutrient intake (g) and total sugar (g), fiber (g) and alcohol (g) by tertiles of energy are presented in Table 2. Table 2 also contains the energy-adjusted macronutrients (as a percentage to energy contribution), and adjusted sugar, fiber and alcohol per 1000 kcal of intake. While unadjusted intakes are statistically different, the adjusted intakes are not statistically different between tertiles of energy. A lower carbohydrate diet score is observed in the first tertile of energy, as compared to other tertiles. A linear trend of the adequacy of intake was observed for proteins, but not for fat or carbohydrates.

In Table 3, the demographic and clinical features are presented by tertiles of low carbohydrate diet scores, with the first tertile indicating higher carbohydrate intake, the second tertile indicating medium carbohydrate intake and the third tertile indicating low carbohydrate intake. Females are more likely than men to report a high carbohydrate diet. The low carbohydrate tertile has significantly higher proportions of higher than adequate fat intake, as compared to medium carbohydrate tertile. Energy-adjusted saturated fat, total fiber and alcohol show significant differences between tertiles of the LCD score. Other clinical characteristics that show differences between tertiles of the LCD score include hemoglobin, uric acid, albumin, total proteins, beta-2 microglobulin, percentage of plasmacytes in the bone marrow and D-dimers. Clinical indicators, namely types of response to therapy and the number of therapeutic lines, are not associated with tertiles of the LCD score. 

Table 4 shows the percentages of nutrition knowledge score per each section and the total score. Compared to the previously published knowledge scores, calculated for the Romanian population [20], Section 1, Section 3 and Section 4 scores were lower, but the score for Section 2 and the total score were similar. The sub-scores per section and the total score did not differ when using demographic and clinical factors (Table 4).

## 4. Discussion

To the best of our knowledge, neither the LCD score nor the nutrition knowledge has been assessed for patients diagnosed with MM. LCD score has been used for several years in relation to the risk of chronic diseases [18,19] and mortality [21]. Nutrition knowledge was previously associated with healthier food choices [22,23,24] and is a necessary, but not a sufficient, component of behavioral change.

By tertiles of energy, the energy-adjusted intake of nutrients did not reach statistical significance, meaning that the increase in energy is not due to a particular macronutrient source of energy. A higher LCD score, associated with a diet lower in carbohydrates, was observed in the lowest tertile of energy (Table 3). Proportions of the adequacy of protein intake showed a linear trend, with the lowest tertile being associated with the lowest proportion of adequacy of intake and reaching up to 100% in the third LCD tertile.

Similar to other populational studies [25], by tertiles of LCD score, women are more likely than men to have higher intakes of carbohydrates. Low carbohydrate diets have become popular due to short-term weight loss, with better results when associated with high protein diets [26,27], but recent research is inconsistent regarding the long-term effects on diabetes [28,29]. High mortality rates and increased risk of chronic diseases were reported among people consuming low carbohydrate diets, especially when low carbohydrate diets were accompanied by high fat intake, especially saturated fat, and a low fiber intake [18,19,21,30,31,32]. In our sample (Table 3), low carbohydrate diets were associated with high saturated fat and alcohol intake and low fiber intake, situating our patients in the high-risk group. Yet, some clinical indicators, such as a significantly lower hemoglobin level and higher prevalence of reported peripheral neuropathy, were reported in our low carbohydrate–high saturated fat tertile.

Then again, high carbohydrate diets are also associated with high morbidity and mortality rates [33,34,35]. As demonstrated by our small sample, compared to the medium tertile of LCD score, which has the highest proportions of macronutrient adequacy (94.4% for carbohydrate, 77.8% for fat and 66.7% for protein), the high carbohydrate tertile, with 52.4% for carbohydrate, 95.2% for fat and 52.4% for protein, has the worst clinical indicators (Table 3). Anemia has a broad implication, i.e., the low hemoglobin and hematocrit, and induces or aggravates hypoxia, impacts the cardiovascular system, is associated with poor quality of life and performance and impairs daily activity. Considering that most MM patients are elderly, the clinical aspects previously presented may be even more important [36]. Anemia is often associated with and aggravated by chronic kidney disease [37].

Several hemostatic and thrombotic anomalies have been reported in patients with MM, with D-dimers being the most commonly reported prothrombotic marker [38]. D-dimers are associated with poor prognosis in patients with cancer [39]. Recent research has shown that hypercalcemia and bone disease are significantly associated with a worse prognosis [40]. The International Staging System, which is based on serum beta-2 microglobulin and albumin levels, is the most widely adopted in multiple myeloma and is also correlated with the prognosis of the disease [40]. Assessment of bone marrow involvement by malignant plasma cells is an important element in the diagnosis and follow-up of patients with multiple myeloma and other plasma cell dyscrasias [40,41].

Patients with MM suffer diagnostic delays due to the complex nature of the disease and non-specificity of symptoms [42,43,44]. Often, due to these delays, patients are diagnosed in the late stage of disease, associated with poor prognosis [40]. In our experience, patients are often diagnosed in stage 3 disease, and this is reflected in our sample by shorter time to follow-up, compared to patients in stages 1 and 2 (Table 1). Patients in disease stage 3 are more likely to report either high carbohydrate intake or low carbohydrate intake, the latter not reaching statistical significance after Bonferroni correction (Table 3). In our sample, patients from the high carbohydrate diet tertile have lower hemoglobin and albumin levels and higher D-dimers, calcium, uric acid, percentage of plasmacytes in the bone marrow and beta-2 microglobulin levels, when compared to patients from the medium carbohydrate tertile. Since our study is cross-sectional, we cannot assume causality, only association. Further research will be able to clarify some of these mechanisms observed between clinical outcomes and patterns of nutrient intake, and the relationships between clinical indicators of disease status.

There are several proposed mechanisms to explain the association of anemia with MM, and the most important seems to be inadequate erythropoietin (EPO) production related to inflammatory cytokines [45] and the high levels of hepcidin, the iron regulatory hormone that works by restricting the iron supply for erythropoiesis [46]. Several dietary factors can aggravate anemia, and phytates, fiber and starches were considered to have inhibitory effects on iron absorption [47]. In a 10-year retrospective study in patients with MM [48], low folate levels were associated with lower levels of hemoglobin. High BMI has a protective effect for anemia, but even in this population, increasing the dietary fat/carbohydrate ratio increases the risk of anemia [49].

Purine-rich foods, high protein intake, alcohol and fructose are known risk factors for increasing the levels of uric acid [50]. Fructose directly regulates uric acid production by increasing ATP degradation to AMP, a uric acid precursor [50]. Intervention studies have shown that the relationship between carbohydrate intake and uric acid production is mediated by the glycemic index of food, with the intake of food with low glycemic index lowering the levels of uric acid [51].

Total nutrition knowledge scores (Table 4) were low, but were similar to those of recent research [20] that used the same instrument. In the sections Expert recommendations, Healthy food choices and Diet, disease and weight associations, our sample scored lower than the general population, so this could represent a starting point in targeted diet interventions. The nutrition knowledge score did not differ by demographics, intake or clinical indicators, though some trends were observed, but did not reach statistical significance.

Patients with MM may gain benefits if they receive nutritional counseling and healthy eating guidance, in order to learn how to sustain a balanced diet, because nutritional needs change during the course of the disease and survivorship [52]. A study by Lee et al. [8] suggests that increased scores in different patterns of intake, such as in the Alternate Healthy Eating Index (AHEI)-2010, Alternate Mediterranean Diet or Dietary Approaches to Stop Hypertension were associated with better prognosis and survival in patients with MM. Intermittent fasting has been proposed as an alternative option in cancer therapy due to autophagy, but has been controversial [53,54,55].

Epigenetic mechanisms via DNA methylation, histone modifications and non-coding RNAs were found to be associated with better evolution and therapeutic response in patients with MM [56]. Although adequate intake of methyl donors such as folate, methionine, choline and vitamins B2, B6, B12 are likely to impact epigenetic mechanisms [57], better understanding of the different polymorphisms from the genes involved in one-carbon metabolism could shed new light on nutrition counseling for patients with MM [58,59].

A major limitation of our study is the small number of participants since recruitment occurred during the second wave of the COVID-19 pandemic. Secondly, participants were all recruited in a single university clinic, thus the results cannot be generalized. A larger multicentric sample would be necessary to be able to generalize these findings. The participants were volunteers, which might have biased selection toward individuals more interested in food, diet and health and with a genuine interest in these matters, as compared to those who declined participation. Since the current study did not receive external funding, not all participants received metabolic panel analyses. Subsequent studies should plan to overcome this limitation.

## 5. Conclusions

Stage 3 disease in patients with MM is associated with either a high carbohydrate or low carbohydrate–high saturated fat diet. Other clinical indicators, besides beta-2 microglobulin, which have been related to poor prognosis, are associated in our sample with a high carbohydrate diet. During disease treatment and survivorship, the nutritional needs of MM patients change, thus a better knowledge of foods and nutritional value could encourage them to make healthier decisions that might extend survival.

## Figures and Tables

**Figure 1 ijerph-18-05444-f001:**
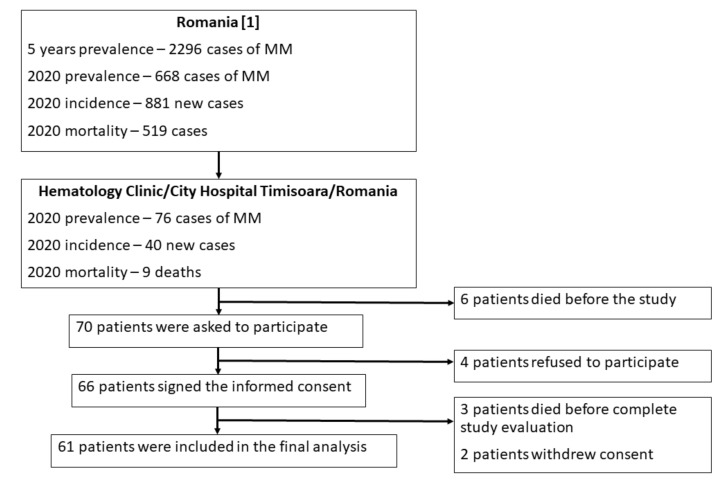
Study flow chart.

**Table 1 ijerph-18-05444-t001:** Clinical features of the cohort of patients with a diagnosis of MM by disease staging (N = 61).

Clinical Features	Measure		Disease Stages 1 and 2 (n1 = 21)	Disease Stage 3 (n2 = 40)	*p*-Value	Total
Sex F	n (%) *		12 (57.1%)	23 (57.5%)	0.979	35 (57.4%)
Age (years)	Mean +/− SD **		65.1 +/− 11.7	65.2 +/− 8.2	0.964	65.2 +/− 9.5
Body mass index (kg/m^2^)	Mean +/− SD **		26.3 +/− 5.1	26.2 +/− 3.7	0.917	26.3 +/− 4.2
Follow-up since diagnosis (months)	Mean +/− SD **		32.9 +/− 19.7	22.1 +/− 14.3	**0.018**	25.8 +/− 17.0
Blood smear alterations	n (%) *		7 (33.3%)	34 (85.0%)	**<0.001**	41 (67.2%)
Anemia	n (%) *		15 (71.4%)	40 (100.0%)	**0.001**	55 (90.2%)
Hemoglobin (g/dl)	Mean +/− SD **		10.9 +/− 1.7	7.3 +/− 0.9	**<0.001**	8.6 +/− 2.1
Hypercalcemia	n (%) *		6 (28.6%)	26 (65.0%)	**0.007**	32 (52.5%)
Serum calcium (mg/dL)	Mean +/− SD **		10.6 +/− 2.3	12.8 +/− 3.1	**0.004**	12.0 +/− 3.0
Alkaline Phosphatase (IU/L)	Mean +/− SD **		71.9 +/− 18.5	116.0 +/− 36.9	**<0.001**	100.8 +/− 38.0
Beta-2 microglobulin ≥ 3.5	n (%) *		13 (61.9%)	40 (100.0%)	**<0.001**	53 (86.9%)
Beta-2 microglobulin (mg/dl)	Mean +/− SD **		3.7 +/− 0.9	11.2 +/− 3.1	**<0.001**	8.6 +/− 4.4
Percentage of plasmacytes > 60%	n (%) *		3 (14.3%)	28 (70.0%)	**<0.001**	31 (50.8%)
Plasmacytes in bone marrow (%)	Mean +/− SD **		36.8 +/− 20.6	68.1 +/− 16.2	**<0.001**	57.3 +/− 23.2
Uric acid (mg/dL)	Mean +/− SD **		5.1 +/− 1.4	7.5 +/− 1.8	**<0.001**	6.7 +/− 2.0
Creatinine (mg/dL)	Mean +/− SD **		1.3 +/− 0.9	2.5 +/− 1.3	**<0.001**	2.1 +/− 1.3
Blood urea nitrogen (mg/dL)	Mean +/− SD **		42.4 +/− 17.3	67.3 +/− 32.3	**<0.001**	58.7 +/− 30.4
Albumin (g/L)	Mean +/− SD **		59.7 +/− 5.7	51.1 +/− 3.9	**<0.001**	54.0 +/− 6.1
Total protein (g/dL)	Mean +/− SD **		6.6 +/− 0.9	9.2 +/− 3.8	**<0.001**	8.3 +/− 3.3
Erythrocyte sedimentation rate (mm/hr)	Mean +/− SD **		87.8 +/− 40.9	114.6 +/− 38.5	**0.014**	105.4 +/− 41.1
C-reactive protein (mg/L)	Mean +/− SD **		8.5 +/− 9.8	21.7 +/− 14.4	**<0.001**	17.2 +/− 14.4
LDH (U/L)	Mean +/− SD **		226.0 +/− 63.0	359.5 +/− 121.8	**<0.001**	313.6 +/− 122.7
Fibrinogen (mg/dL)	Mean +/− SD **		421.4 +/− 123.6	507.5 +/− 152.6	**0.030**	477.8 +/− 148.1
D-dimers (ng/mL)	Mean +/− SD **		220.7 +/− 59.7	340.4 +/− 118.4	**<0.001**	299.1 +/− 116.5
Serum-free light chain	Kappa	n (%) *	16 (76.2%)	26 (65.0%)	**0.370**	42 (68.9%)
	Lambda		5 (23.8%)	14 (35.0%)		19 (31.1%)
Immunoglobulin type	IgG	n (%) *	15 (71.4%)	23 (57.5%)	**0.429**	38 (62.3%)
	IgA		5 (23.8%)	16 (40.0%)		21 (34.4%)
	IgM		1 (4.8%)	1 (2.5%)		2 (3.3%)
Infections	n (%) *		2 (9.5%)	7 (17.5%)	**0.479**	9 (14.8%)
Myelosuppression	n (%) *		0 (0.0%)	10 (25.0%)	**0.011**	10 (16.4%)
Chronic kidney disease	n (%) *		3 (14.3%)	21 (52.5%)	**0.004**	24 (39.3%)
Peripheral neuropathy	n (%) *		0 (0.0%)	31 (77.5%)	**<0.001**	31 (50.8%)
Osteoporosis	n (%) *		1 (4.8%)	17 (42.5%)	**0.002**	18 (29.5%)
Depression	n (%) *		14 (66.7%)	19 (47.5%)	**0.153**	33 (54.1%)
Constipation/diarrhea	n (%) *		0 (0.0%)	16 (40.0%)	**0.001**	16 (26.2%)
Autologous stem cell transplantation	n (%) *		0 (0.0%)	9 (22.5%)	**0.021**	9 (14.8%)
Response	Partial remission	n (%) *	8 (38.1%)	15 (37.5%)	**0.370**	23 (37.7%)
	Total remission		9 (42.9%)	10 (25.0%)		19 (31.1%)
	Stable disease		3 (14.3%)	9 (22.5%)		12 (19.7%)
	Progressive disease		1 (4.8%)	6 (15.0%)		7 (11.5%)
Therapeutic lines administered	Mean +/− SD ***		3.2 +/− 1.4	2.8 +/− 1.6	**0.247**	2.9 +/− 1.5

Notes: Data are presented as mean +/− SD, or as n (%), as appropriate. * Chi-square test, ** independent samples *t*-test, *** Mann–Whitey test. *p*-values in bold are statistically significant. Abbreviations: SD = standard deviation, n = number of participants.

**Table 2 ijerph-18-05444-t002:** Nutrient intake per tertile of energy intake (N = 61).

Nutrient Intake		Measure	Tertiles of Energy (kcal) Intake			*p*-Value for Trend
			First (n1 = 20)	Second (n2 = 21)	Third (n3 = 20)	
Energy (kcal)		Mean +/− SD *	797.6 +/− 139.2	1259.1 +/− 164.0	2296.0 +/− 1030.9	**<0.001**
Protein (g)		Mean +/− SD *	39.7 +/− 9.1	54.8 +/− 11.8	92.9 +/− 34.9	**<0.001**
Fat total (g)		Mean +/− SD *	31.1 +/− 6.9	44.8 +/− 11.5	80.9 +/− 34.3	**<0.001**
Carbohydrate (g)		Mean +/− SD *	89.4 +/− 20.2	161.6 +/− 36.8	301.5 +/− 188.4	**<0.001**
Saturated fat (g)		Mean +/− SD *	10.1 +/− 2.2	15.7 +/− 4.8	27.0 +/− 9.9	**<0.001**
Fiber total dietary (g)		Mean +/− SD *	6.9 +/− 2.2	13.8 +/− 6.5	25.0 +/− 25.7	**0.002**
Alcohol (g)		Mean +/− SD *	1.2 +/− 2.0	1.2 +/− 1.8	3.1 +/− 3.6	**0.036**
Percentage of energy from fat		Mean +/− SD *	35.0 +/− 4.7	31.9 +/− 6.3	32.7 +/− 9.3	0.361
Percentage of energy from carbohydrates		Mean +/− SD *	44.8 +/− 6.6	51.3 +/− 9.3	50.8 +/− 12.3	0.057
Saturated fat (g)/1000 kcal		Mean +/− SD *	1.3 +/− 0.1	1.2 +/− 0.3	1.2 +/− 0.4	0.946
Total fiber (g)/1000 kcal		Mean +/− SD *	8.6 +/− 2.5	10.9 +/− 4.7	9.9 +/− 5.3	0.237
Alcohol (g)/1000 kcal		Mean +/− SD *	1.5 +/− 2.7	1.0 +/− 1.4	1.4 +/− 1.8	0.648
LCD score		Mean +/− SD **	19.9 +/− 5.8	14.7 +/− 7.4	15.0 +/− 8.3	**0.045**
Protein adequate intake n (%)		n (%) **	7 (35.0%)	14 (66.7%)	20 (100.0%)	**<0.001**
Fat adequate intake n (%)		n (%) **	10 (50.0%)	16 (76.2%)	11 (55.0%)	0.193
Carbohydrate intake	Below adequate n (%)	n (%) **	9 (45.0%)	6 (28.6%)	7 (35.0%)	0.185
	Adequate n (%)	n (%) **	11 (55.0%)	11 (52.4%)	7 (35.0%)	
	Over adequate n (%)	n (%) **	0 (0.0%)	4 (19.0%)	6 (30.0%)	

Notes: Data are presented as mean +/− SD, or as n (%), as appropriate. * ANOVA, ** Kruskall–Wallis test; *p*-values in bold are statistically significant. Abbreviations: SD = standard deviation, n = number of participants, LCD = low carbohydrate diet.

**Table 3 ijerph-18-05444-t003:** Clinical and intake variables per tertile of LCD score.

Clinical and Intake Variables		Measure	Tertiles Of LCD Score			*p*−Values
			First (High Carb Diet) n1 = 21	Second (Medium Carb Diet) n2 = 18	Third (Low Carb Diet) n3 = 22	
Sex	M	n (%) *	6 (28.6%)	5 (27.8%)	15 (68.2%)	**0.011**
	F		15 (71.4%)	13 (72.2%)	7 (31.8%)	
Age		Mean +/− SD **	66.3 +/− 7.3	61.1 +/− 11.2	67.4 +/− 9.1	0.087
Follow-up since diagnosis (months)		Mean +/− SD **	25.0 +/− 17.7	29.6 +/− 16.3	23.5 +/− 17.1	0.524
Disease stage	1 and 2	n (%) *	2 (9.5%)	12 (66.7%)	7 (31.8%)	**0.001**
	3		19 (90.5%) ^a^	6 (33.3%)	15 (68.2%)	
Adequate fat intake	Increased	n (%) *	1 (4.8%)	4 (22.2%)	19 (86.4%)	**<0.001**
	Yes		20 (95.2%)	14 (77.8%)	3 (13.6%) ^a^	
Adequate carbohydrate intake	Decreased	n (%) *	0 (0.0%)	1 (5.6%)	21 (95.5%)	**<0.001**
	Yes		11 (52.4%)	17 (94.4%)	1 (4.5%)	
	Increased		10 (47.6%)	0 (0.0%)	0 (0.0%)	
Adequate protein intake	Decreased	n (%) *	10 (47.6%)	6 (33.3%)	4 (18.2%)	0.125
	Yes		11 (52.4%)	12 (66.7%)	18 (81.8%)	
Saturated fat (g)/1000 kcal		Mean +/− SD *	1.0 +/− 0.2 ^a^	1.2 +/− 0.2	1.5 +/− 0.2 ^a^	**<0.001**
Total fiber (g)/1000 kcal		Mean +/− SD *	13.0 +/− 5.4 ^a^	9.8 +/− 2.5	6.8 +/− 1.6 ^a^	**<0.001**
Alcohol (g)/1000 kcal		Mean +/− SD *	0.7 +/− 1.2	0.8 +/− 0.8	2.3 +/− 2.8 ^a^	**0.013**
Hemoglobin (g/dl)		Mean +/− SD **	8.1 +/− 1.7 ^a^	9.8 +/− 2.2	8.0 +/− 2.0 ^a^	**0.010**
Alkaline phosphatase (IU/L)		Mean +/− SD **	114.4 +/− 31.9	88.1 +/− 35.8	98.1 +/− 42.2	0.088
Erythrocyte sedimentation rate (mm/hr)		Mean +/− SD **	104.7 +/− 46.3	102.8 +/− 40.9	108.1 +/− 37.5	0.919
Serum calcium (mg/dL)		Mean +/− SD **	12.1 +/− 3.3	11.8 +/− 2.7	12.2 +/− 3.1	0.928
Uric acid (mg/dL)		Mean +/− SD **	7.7 +/− 1.8 ^a^	5.9 +/− 2.2	6.3 +/− 1.7	**0.006**
Creatinine (mg/dL)		Mean +/− SD **	2.2 +/− 0.9	1.6 +/− 1.2	2.4 +/− 1.5	0.131
Blood urea nitrogen (mg/dL)		Mean +/− SD **	58.7 +/− 21.2	46.9 +/− 21.0	68.4 +/− 40.4	0.083
Albumin (g/L)		Mean +/− SD **	51.3 +/− 4.1 ^a^	56.8 +/− 7.6	54.1 +/− 5.6	**0.018**
Total proteins (g/dL)		Mean +/− SD **	9.9 +/− 3.6 ^a^	7.2 +/− 2.4	7.6 +/− 3.2	**0.018**
Beta-2 microglobulin (mg/dl)		Mean +/− SD **	10.7 +/− 4.0 ^a^	5.9 +/− 3.4	8.8 +/− 4.5	**0.002**
C-reactive protein (mg/L)		Mean +/− SD **	20.3 +/− 17.7	12.0 +/− 9.7	18.3 +/− 13.5	0.179
Plasmacytes in bone marrow (%)		Mean +/− SD **	67.5 +/− 18.5 ^a^	44.4 +/− 23.5	58.2 +/− 22.7	**0.006**
LDH (U/L)		Mean +/− SD **	300.7 +/− 104.2	276.8 +/− 107.3	355.9 +/− 142.0	0.106
Fibrinogen (mg/dL)		Mean +/− SD **	503.1 +/− 139.9	453.5 +/− 149.4	473.5 +/− 157.4	0.580
D-dimers (ng/mL)		Mean +/− SD **	346.0 +/− 122.6 ^a^	252.4 +/− 114.3	292.6 +/− 99.0	**0.039**
Response n (%)	Partial remission	n (%) ***	9 (42.9%)	8 (44.4%)	6 (27.3%)	0.674
	Total remission		4 (19.0%)	6 (33.3%)	9 (40.9%)	
	Stabile disease		5 (23.8%)	3 (16.7%)	4 (18.2%)	
	Progressive disease		3 (14.3%)	1 (5.6%)	3 (13.6%)	
Infections		n (%) *	6 (28.6%)	1 (5.6%)	2 (9.1%)	0.087
Myelosuppression		n (%) *	6 (28.6%)	1 (5.6%)	3 (13.6%)	0.144
Chronic kidney disease		n (%) *	8 (38.1%)	7 (38.9%)	9 (40.9%)	0.982
Peripheral neuropathy		n (%) *	16 (76.2%) ^a^	3 (16.7%)	12 (54.5%) ^a^	**0.001**
Osteoporosis		n (%) *	7 (33.3%)	4 (22.2%)	7 (31.8%)	0.722
Depression		n (%) *	12 (57.1%)	10 (55.6%)	11 (50.0%)	0.888
Constipation/diarrhea		n (%) *	7 (33.3%)	1 (5.6%)	8 (36.4%)	0.061
Therapeutic lines		Mean +/− SD *	2.76 +/− 1.7	3.22 +/− 1.4	2.86 +/− 1.5	0.628

Notes: Data are presented as mean +/- SD, or as n (%), as appropriate. * Kruskal–Wallis test; ** ANOVA; *** chi-square test. ^a^ Superscript letter denotes a significant difference, as compared to the second tertile using Mann–Whitney test with Bonferroni adjustment, or post hoc tests with Sidak adjustment, as appropriate. *p*-values in bold are statistically significant. Abbreviations: SD = standard deviation, n = number of participants, LCD = low carbohydrate diet.

**Table 4 ijerph-18-05444-t004:** Nutrition knowledge score percentages per sections and total score (N = 61 participants).

Factors		Section 1 Achievement (%)Expert Recommendations	Section 2 Achievement (%)Food Groups	Section 3 Achievement (%) Healthy Food Choices	Section 4 Achievement Diet, Disease and Weight Associations	Total Score Achievement (%)
Median achievement		64.0 +/− 9.4 *	63.5 +/− 9.2	71.6 +/− 13.9 *	65.0 +/− 11.2 *	65.2 +/− 7.1
Sex	M	63.7 +/− 10.3	63.1 +/− 8.4	71.6 +/− 16.2	67.2 +/− 9.3	65.5 +/− 7.7
	F	64.3 +/− 8.8	63.7 +/− 9.8	71.6 +/− 12.2	63.4 +/− 12.3	64.9 +/− 6.8
Age category	≤65 years	65.9 +/− 10.4	63.8 +/− 10.5	73.4 +/− 13.6	67.3 +/− 11.0	66.5 +/− 7.5
	>65 years	62.0 +/− 7.9	63.1 +/− 7.8	69.7 +/− 14.3	62.7 +/− 11.1	63.8 +/− 6.6
Education	High school or less	61.8 +/− 10.2	63.9 +/− 9.0	73.6 +/− 14.6	65.9 +/− 10.1	65.4 +/− 7.4
	At least college degree	66.5 +/− 7.9	63.0 +/− 9.5	69.5 +/− 13.1	64.0 +/− 12.4	64.9 +/− 7.0
Living with underage individuals	Yes	64.3 +/− 9.3	63.7 +/− 9.4	71.0 +/− 13.6	65.0 +/− 11.2	65.2 +/− 7.4
	No	61.1 +/− 10.5	61.6 +/− 6.4	76.9 +/− 16.9	65.1 +/− 12.3	64.6 +/− 4.9
Percentile group of LCD score	First (high carb)	62.4 +/− 9.1	62.8 +/− 8.0	73.3 +/− 12.6	61.9 +/− 12.1	64.1 +/− 5.2
	Second (medium)	67.0 +/− 8.8	63.1 +/− 11.7	68.8 +/− 13.6	65.3 +/− 11.4	65.3 +/− 8.6
	Third (Low carb)	63.1 +/− 9.9	64.4 +/− 8.2	72.4 +/− 15.7	67.7 +/− 9.7	66.1 +/− 7.6
Anemia	No	66.7 +/− 7.0	60.2 +/− 13.1	70.5 +/− 13.2	61.9 +/− 6.0	63.4 +/− 7.8
	Yes	63.7 +/− 9.6	63.8 +/− 8.7	71.7 +/− 14.1	65.4 +/− 11.6	65.4 +/− 7.1
Hypercalcemia	No	65.5 +/− 6.4	62.9 +/− 9.4	73.2 +/− 13.6	63.7 +/− 9.0	65.2 +/− 6.8
	Yes	62.7 +/− 11.4	64.0 +/− 9.0	70.2 +/− 14.3	66.2 +/− 12.9	65.2 +/− 7.6
Chronic kidney disease	No	65.3 +/− 8.3	63.7 +/− 9.4	73.0 +/− 13.3	66.8 +/− 10.2	66.2 +/− 6.8
	Yes	62.0 +/− 10.7	63.1 +/− 9.0	69.6 +/− 15.0	62.3 +/− 12.3	63.6 +/− 7.5
Infections	No	65.2 +/− 8.9	64.4 +/− 9.1	72.0 +/− 13.6	65.3 +/− 11.5	65.9 +/− 7.2
	Yes	57.4 +/− 9.6	58.3 +/− 8.4	69.2 +/− 16.3	63.5 +/− 9.5	61.0 +/− 5.0

**Notes:** Data is presented as mean +/− SD. * Statistically significant lower achievement score, when compared to the population median, with the following values: Section 1—66.7%, Section 2—63.8%, Section 3—76.9%, Section 4—71.4%, total score—65.9% [20]; between factors comparisons with independent samples *t*-test or ANOVA, as appropriate.

## Data Availability

Raw data for variables of the cohort are available on request.

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
