# Peer review of "High Carbohydrate Diet Is Associated with Severe Clinical Indicators, but Not with Nutrition Knowledge Score in Patients with Multiple Myeloma"

_ijerph, 2021, doi:10.3390/ijerph18105444_

Round 1
Reviewer 1 Report
The manuscript describes a cohort study of 61 patients diagnosed with multiple myeloma. The aim of the study is to evaluate the eating habits and nutritional awareness of patients, identifying correlations with clinical parameters. Individuals in the first and third tertile of carbohydrate consumption show significant correlations with some worse clinical values. The nutritional knowledge outcomes appear to be in some cases worse than the literature data on the general population. The authors recommend evaluating the nutritional aspects in patients with multiple myeloma for the possible risk of clinical alterations related to macronutrient imbalances, especially in the case of high carbohydrate intake.
The manuscript focuses on the possible influence of nutrition on the management of patients with multiple myeloma, although it is not a condition previously related to eating habits compared to other cancers.
Unfortunately, the sample is very small and the study design does not allow to identify the causal mechanisms that relate nutrition with the clinical indices evaluated.
This raises concerns about the relevance and innovation of the results obtained.
Possible mechanisms should be proposed that explains the identified correlations, especially since there does not seem to be a clear logical involvement of carbohydrate intake on hemoglobin levels, stage of disease, uric acid levels, bone marrow plasmacytes, etc.
It would be better to describe in a flow chart the individuals included, those who refused to participate and those excluded, specifying the reasons for exclusion. How many individuals has it been proposed to? Was the study offered to all patients of the follow-up in the reporting period? Furthermore, at the end of the manuscript, there is mention of voluntary enrollment. How was participation proposed?
Why were metabolic indices such as glycemia, insulin, cholesterol, which would have been more consistent with the nutritional survey, not evaluated?
Minor aspects:
- There are bold terms along the text that shouldn't be, such as "was" on line 102; “14]” on line 107 and more generally many of the numbered references along the text.
- In line 21 replace "nutrient intake" with "diet"
- The authors state that 57.4% of the participants are overweight or obese. In this context, it is inconsistent with the statement on line 262-263 in which the authors state that patients with MM are at risk of weight loss. In the reference sample, there are no underweight individuals, if I understand correctly.
- In table 2 it seems to me to make no sense to evaluate the correlation between energy intake and tertiles of energy intake. What should it highlight?
I suggest that the authors do a linguistic check to make the reading smoother.
Reviewer 2 Report
In the present clinical study, Borsi and coworkers assessed the nutrition knowledge and diet quality in a group of patients with a diagnosis of multiple myeloma (MM). Specifically, anthropometric, clinical and biological, skeletal survey evaluations along with nutritional intake and general nutrition knowledge were performed on 61 patients with a current diagnosis of multiple myeloma. The authors showed that Stage 3 disease in patients with MM is associated with either high carbohydrate or low carbohydrate high saturated fat. Other clinical indicators, besides beta 2-microglobulin, which have been related to poor prognosis, are associated, in this sample with a high carbohydrate diet. Consequently, the authors stated that during disease treatment and survivorship, the nutritional needs of MM patients are changing, and a better knowledge of foods and nutritional value could encourage them to make healthier decisions that might extend survival. Overall, I think that the paper fits within the scope of this journal and the data are interesting even if future randomized, double-blind, controlled studies about the association between diet and MM indicators are warranted. I make some suggestions for improve the overall quality of paper.
- The authors should assess the sample size and need to do a power calculation. It is a major point to address in this study.
- Based on these results, it is possible, in the author's opinion, obtain benefit through nutraceuticals and/or functional foods? Please make an appropriate comment in discussion section of manuscript and insert appropriate references.
- Please focus the discussion on the potential positive effects of Mediterranean Style diet and Oriental Diet in terms of prevention and therapy of MM.
- An intermittent fasting diet might be beneficial to MM patients? Please discuss this intriguing topic.
- Several studies suggest that numerous nutritional compounds have epigenetic targets in MM. Please better clarify this aspect in light of the results reported in this clinical study and improve the discussion section, accordingly.
Reviewer 3 Report
Thank you for asking me to review this manuscript.
This is an interesting paper on an important topic; however, I would like to make some suggestions on how to make the paper stronger. Please see my comments below.
- Please use the Microsoft Word template for IJERPH (use the words Background: Methods: Results and Conclusions). The abstract will be more readable.
- Please add a short paragraph on the definition BMI in the method. Did you use WHO BMI Criteria?
- The paper is very fluent, clear and easy to read. Related work is critically analyzed.
Round 2
Reviewer 1 Report
The authors responded sufficiently to requests and improved the manuscript accordingly. Thank you for your revisions
Reviewer 2 Report
The authors have satisfactorily responded to all my questions and made the necessary changes to the manuscript.